# Analysis of the Downscaling Effect and Definition of the Process Fingerprints in Micro Injection of Spiral Geometries

**DOI:** 10.3390/mi10050335

**Published:** 2019-05-22

**Authors:** Antonio Luca, Oltmann Riemer

**Affiliations:** Laboratory for Precision Machining (LFM), Leibniz Institute for Materials Engineering—IWT, University of Bremen, Badgasteiner Straße 2, 28359 Bremen, Germany; riemer@iwt.uni-bremen.de

**Keywords:** microinjection moulding, process fingerprints, flow length, quality assurance

## Abstract

Microinjection moulding has been developed to fulfil the needs of mass production of micro components in different fields. A challenge of this technology lies in the downscaling of micro components, which leads to faster solidification of the polymeric material and a narrower process window. Moreover, the small cavity dimensions represent a limit for process monitoring due to the inability to install in-cavity sensors. Therefore, new solutions must be found. In this study, the downscaling effect was investigated by means of three spiral geometries with different cross sections, considering the achievable flow length as a response variable. Process indicators, called “process fingerprints”, were defined to monitor the process in-line. In the first stage, a relationship between the achievable flow length and the process parameters, as well as between the process fingerprints and the process parameters, was established. Subsequently, a correlation analysis was carried out to find the process indicators that are mostly related to the achievable flow length.

## 1. Introduction

The application of small components with dimensions in the micrometre and nanometre ranges has largely increased in various engineering fields over the recent decades [1]. Micro components are gaining importance in areas such as health care, optical products, automotive industry, communication, biotechnology and so forth. The demand for micro parts made of thermoplastic polymers is becoming increasingly widespread due to the need for reduced weight, high chemical resistance, low production costs and ease of fabrication—even in complex shapes. Therefore, advanced micro manufacturing technologies are fundamental to support their production. In this context, most of these products are nowadays manufactured by microinjection moulding. This process can be conceived of as a miniaturised variant of the conventional injection moulding process, with the intention of combining high productivity with the capability to manufacture micro components [2]. A substantial difference [3] between the conventional and the micro process is the filling of the cavity, which becomes much more challenging at the microscale. The downscaling of micro components leads to a relatively large surface-to-volume ratio and, consequently, to an increased heat flux at the mould–melt interface. Hence, solidification occurs quickly, hindering the complete filling of the cavity [4]. High levels of injection speed as well as melt and mould temperatures are typically required in order to favour the replication capability of the process [5,6,7]. However, the process window becomes narrower at the microscale, as the levels of the injection moulding process parameters are largely limited by the existence of polymer degradation. Therefore, process optimisation is a fundamental step for manufacturing products that comply with design specifications. When the geometric characteristics of the components are the response variables, experimental investigations based on off-line digital measurements are fundamental to tune the process, making process optimisation a time-consuming task. Additionally, high throughput rates do not allow for measurement of all the produced micro components in three dimensions [8]. A solution to this problem is the application of in-line monitoring techniques to the microinjection moulding process as a tool for quality assurance.

For example, in-line measurements can be carried out to monitor cavity pressure. This factor is in most studies identified [9,10,11] as the process variable that best outlines the evolution of the moulding cycle. However, in the micro process, the cavity dimensions can be comparable or even larger than that of typical sensors, therefore interfering with their use [12]. A solution to this issue is to monitor the hydraulic pressure provided by the injection plunger or screw, since this quantity can be extracted from the machine data for each moulding cycle without the need to use any further sensor. The limitation of this method is the difference between hydraulic and cavity pressures, which means that the pressure measured at the screw is not representative of the behaviour of the polymer melt inside the cavity.

The selection of the main injection moulding process parameters has a relevant impact on the recorded process indicators (e.g., cavity pressure [13], injection speed [14], demoulding forces [15], etc.). Thus, the process can be successfully monitored by controlling these variables. Nevertheless, a relationship between those indicators, called “process fingerprints”, and the dimensional quality of the produced micro components has not been established so far. In this investigation, a correlation between the flow length (the response) and the monitored process fingerprints was achieved. Thus, quality assurance can be performed in-line by only controlling their values, enhancing its robustness and reducing the quality control time. 

## 2. Materials and Methods 

### 2.1. Spiral Geometry

In this work, micro components having the same spiral geometry but different cross-section dimensions were investigated. The aim was to evaluate the downscaling effect on the filling behaviour and the process conditions by measuring the manufactured parts and monitoring representative process variables. In this case, the geometric characteristic of the component considered as a response variable was represented by the achievable flow length of the polymeric material, which depended on the selected process parameters and the process conditions. 

The three selected flow spiral geometries had rectangular cross sections of 1 *×* 1 mm^2^, 0.5 × 0.5 mm^2^ and 0.25 × 0.25 mm^2^ (see Figure 1).

The cavities were designed in a changeable mould unit and integrated a central ejector (see Figure 2). The tooling process was carried out on a five-axis milling centre (DMG Sauer Ultrasonic 20 Linear).

### 2.2. Measurement Method

The flow length was determined by recording the solidified and demoulded part with a VHX-6000 digital microscope from the manufacturer Keyence. For this purpose, the flow spirals were divided into quadrants (see Figure 3). 

Each quadrant had a larger radius than the previous one. With the straight flow length from the sprue to the quadrant 0 and the different radii, the maximum flow length could be calculated up to the end of each quarter circle. For example, the achievable flow length of the spiral with a cross section of 1 × 1 mm^2^ for different radii is illustrated in Table 1.

If the last quadrant was not filled completely, a filling ratio of the incompletely filled quadrant was calculated according to the following equation:l = r · π · α/180°.(1)

As an example, in Figure 4, the measurements of the angle α for three different spiral parts (with cross sections of 1 × 1, 0.5 × 0.5 and 0.25 × 0.25 mm^2^, respectively) are illustrated.

### 2.3. Microinjection Moulding Machine

Injection moulding experiments were carried out using a microinjection moulding machine (Desma FormicaPlast 2K) consisting of a two-phase piston injection unit and a pneumatic injection drive (see Figure 5). The first phase refers to a heated plasticisation zone with a vertically positioned plasticising piston, while the second phase to a horizontally positioned piston for precision injection that has far more accurate control over the injected polymer melt than large diameter screws.

### 2.4. Polymeric Material

The material chosen to perform the experiments was uncoloured POM N23200035, a thermoplastic polymer with an extremely low coefficient of friction and sliding wear when mated with smooth metal surfaces. Its main properties are listed in Table 2.

### 2.5. Design of Experiments (DoE)

DoE is a standardised approach to determine the relationship between factors affecting a process and its output. Particularly, it helps to identify the critical factors affecting the desired output, thus enabling optimisation of the entire process. The parameters selected for this investigation were: injection speed (Vinj), melt temperature (Tmelt), mould temperature (Tmould) and holding pressure (Phold). The values were based on the material recommendations and preliminary experiments. Two levels of each parameter were chosen. These values were chosen in such a way that they covered a wide range of variations for each of the selected process parameters (see Table 3). 

In order to avoid excessive flash formation, the holding pressure was set at relatively low values. The velocity/pressure switch-over point was set at the packing pressure value. A four-factor full-factorial design consisting of 16 experiments was carried out (see Table 4). For statistical assurance, 10 moulded parts were moulded for every set of process conditions. In order to reach a steady state when changing from one experiment to another, the first seven test specimens were discarded and the following three were kept for evaluation.

### 2.6. Process Monitoring

Pressure and velocity of the injection plunger were recorded in-line during the injection moulding cycle. No external sensor was used, as these process variables were derived from the machine data. These data are available to any machine user and easy to access. The injection pressure was recorded via a strain gauge transducer mounted on the back of the injection plunger, while the injection speed was acquired via the speed of the motor driving the plunger through the control unit of the machine. The recorded speed and pressure needed no alignment with respect to the timescale, as they were acquired synchronously.

The dependence of pressure and velocity on the injection moulding process parameters was investigated by identifying some variables referred to as process fingerprints. These process indicators well characterise the pressure and velocity curves and are defined as follows: Maximum injection pressure, Pmax: this value is defined as the maximum injection pressure recorded during each moulding cycle. This indicator is related to the filling behaviour of the cavity, since the pressure peak is increased by the small size of the channels.Mean injection pressure, Pmean: this quantity is calculated as the average of the pressure values recorded from the start to the end of the moulding cycle.Mean injection speed, Vmean: the mean injection speed is defined as the average velocity that characterises the filling phase. The speed values are recorded in the time interval between the start of the acceleration of the injection plunger and when it stops at the switch-over point.

## 3. Results and Discussion

### 3.1. Achievable Flow Length

Figure 6 shows the results of the achievable flow lengths for flow spirals with different cross sections (1 × 1, 0.5 × 0.5 and 0.25 × 0.25 mm^2^) and different injection moulding process parameters. The error bars represent the standard deviations for the three parts taken into account for each experiment. Higher flow length values were achieved for higher levels of the process parameters. In the case of experiment 1, the achieved flow length for the spiral with a cross section of 0.5 × 0.5 mm^2^ was slightly higher than the one with the cross section of 1 × 1 mm^2^. This was an exception, since for all other experiments, the flow length increased with the size of the cross sections. For all other experiments, the different spirals showed similar behaviour. Comparing the results for cross sections of 1 × 1 and 0.5 × 0.5 mm^2^, the difference in flow length could be estimated, in most cases, with a factor of two in favour of the larger cross section. Considering the results for cross sections of 0.5 × 0.5 and 0.25 × 0.25 mm^2^, this difference was more evident. In this case, a factor of three in favour of the cross section of 0.5 × 0.5 mm^2^ can be observed. Therefore, the downscaling of the cross section does not give flow lengths exactly proportional to the size of the cross section, but it is possible to find a factor representative of this behaviour.

The results from the experimental campaign were analysed considering the flow length as the response variable. An ANOVA with a main effect plot for identifying the factors that significantly affect the response was also employed. The statistical analysis was carried out using the statistical software Minitab 18. In Figure 7, the results for the three spirals are illustrated. 

Figure 7a,c show a similar effect of the process parameters on the achievable flow length. In this case, the melt temperature was the most significant parameter. In the case of a 0.5 × 0.5 mm^2^ cross section, the factors gave a substantially different response. The injection speed was the parameter that had the highest influence, followed by mould temperature, melt temperature and holding pressure. Furthermore, higher flow lengths were recorded for a holding pressure of 400 bar, unlike the other two cases, where higher flow lengths were obtained for a holding pressure of 200 bar. However, this difference can be neglected, since the holding pressure is the parameter that has less of an influence on the response. 

### 3.2. Process Fingerprint Analysis

The process fingerprint analysis was carried out to identify the sensitivity of these process indicators with respect to the process parameters. This dependency is shown in the following diagrams (Figure 8, Figure 9 and Figure 10). For each spiral geometry, the values of maximum injection pressure, mean injection pressure and mean injection speed were obtained during the experiments. Subsequently, an average of these values for the three spirals was carried out, considering experiments with the same process parameters. 

Figure 8 shows the results for the maximum injection pressure. This indicator mostly depended on the selected injection speed and holding pressure values, as an increase in both parameters resulted in an increase of this process indicator. The influence of the holding pressure was due to the fact that the machine was set to switch from the filling to the holding phase at a given pressure value. In this case, the selection of a higher holding pressure implies that the injection pressure rises more before switching to the holding profile. The injection speed influenced the maximum injection pressure because a higher pressure requires a fluid flowing at a higher speed. 

Figure 9 shows the results for the average injection speed. As can be observed, its value was predominantly influenced by the holding pressure and injection speed. The effect of the injection speed was obvious, due to the setting of a higher injection velocity. In the case of holding pressure, a higher level of this process parameter required a lower deceleration of the injection plunger. Therefore, a higher mean injection speed was observed during the moulding cycle. On the other side, melt and mould temperature had a slight effect. 

Figure 10 shows the results for the mean injection pressure. This indicator was mostly influenced by the holding pressure. This was because the mean pressure was calculated considering the entire moulding cycle, which was dominated above all by the holding phase. The other parameters had a definitely lower influence on the mean injection pressure.

### 3.3. Correlation Analysis

A correlation analysis was carried out to establish an efficient quality control based only on the measurement of the process fingerprints. Thus, the achievable flow length was correlated with the process fingerprints (maximum injection pressure, mean injection pressure and mean injection speed). The coefficient of correlation was calculated with the following equation:(2)$$r\left(x,\;y\right)=\frac{\sum_{i}[\left(X_{i}-\bar{X}\right)\left(Y_{i}-\bar{Y}\right)]}{\sqrt{\sum_{i}[{\left(X_{i}-\bar{X}\right)}^{2}\cdot \sum_{i}[{\left(Y_{i}-\bar{Y}\right)}^{2}}}$$
where $\bar{X}$ and $\bar{Y}$ are the mean values of flow length and process fingerprints, respectively, and $X_{i}$ and $Y_{i}$ are their values obtained during the experiments. The coefficient of correlation can vary between +1 and −1. A value of 0 indicates that there is no correlation between the flow length and the process fingerprints, while a value of +1 or −1 indicates a perfect positive or negative correlation respectively. As the values of the process fingerprints for each experiment were an average of three values recorded for each of the three flow spirals, we also considered a single value of flow length for each experiment, making an average of the three values. The following Figure 11 shows the results of the correlation analysis: 

The results of the correlation analysis showed that the mean injection speed was the process fingerprint that had the strongest correlation with the achievable flow length, followed by maximum injection pressure and mean injection pressure. While the maximum injection pressure had a degree of correlation similar to the mean injection speed, the mean injection pressure showed a significantly lower correlation value, meaning that this process fingerprint is not useful to establish quality assurance based on this process indicator. On the other side, mean injection speed and maximum injection pressure act as a link between process monitoring and achievable flow length, representing useful process indicators for faster quality assurance of the injection-moulded micro spiral parts. By using this method, it is possible to have an estimation of the achievable flow length by analysing the process fingerprint values.

## 4. Conclusions

The aim of this research was dual: on one hand, the downscaling effect of spiral components with the same geometries but different cross sections was investigated, using the achievable flow length as the response variable; on the other hand, an optimisation method to reduce the off-line inspection effort of the moulded micro components was established. We demonstrated that the downscaling of the cross sections did not have a linear relationship with the achievable flow lengths. It was possible to identify a factor of two between the flow length achieved with cross sections 1 × 1 mm^2^ and 0.5 × 0.5 mm^2^ and a factor of three between the achieved flow length with cross sections of 0.5 × 0.5 mm^2^ and 0.25 × 0.25 mm^2^. A process in-line monitoring technique was implemented, finding a relationship between the process fingerprints and the microinjection moulding process parameters. Finally, a correlation analysis was carried out to relate the response (i.e., achievable flow length) with the process fingerprints. By monitoring these process indicators, it will be possible to predict the response, thus avoiding time-consuming off-line measurements and performing in-line quality assurance. This developed method has the potential to be applied to other micro components by defining their process fingerprints and responses (e.g., diameters, height of specific features, etc.) depending on the geometry of the micro components being investigated. 

## Figures and Tables

**Figure 1 micromachines-10-00335-f001:**
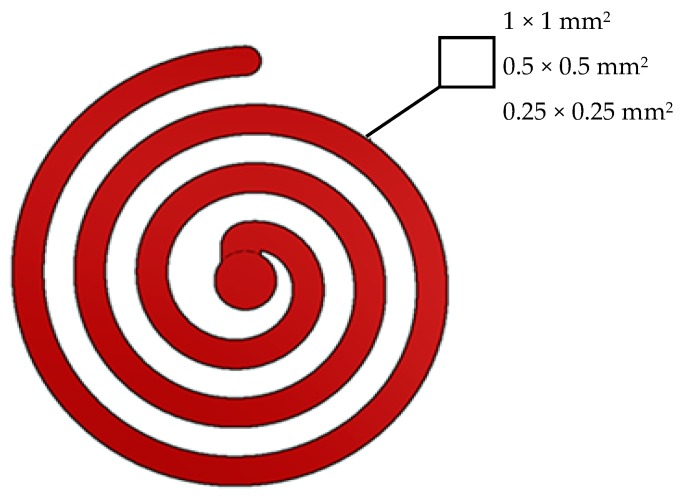
Spirals with different cross sections.

**Figure 2 micromachines-10-00335-f002:**
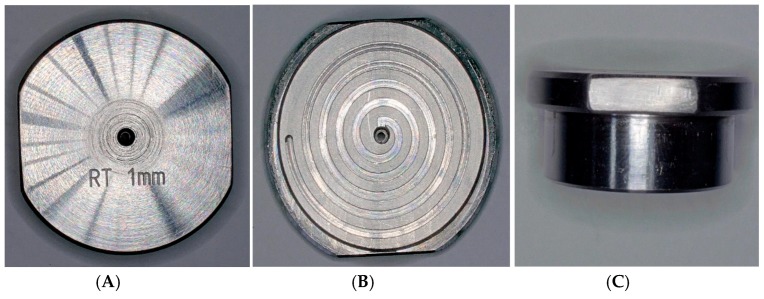
Changeable mould unit including a central ejector for spiral geometries with a cross section of 1 × 1 mm^2^. (**A**) Bottom view; (**B**) top view; (**C**) side view.

**Figure 3 micromachines-10-00335-f003:**
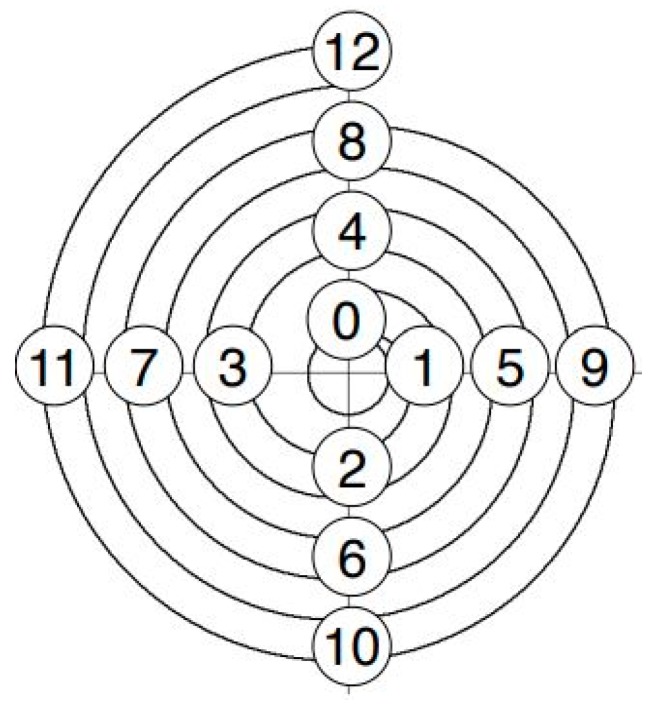
Quadrants of the flow spiral.

**Figure 4 micromachines-10-00335-f004:**
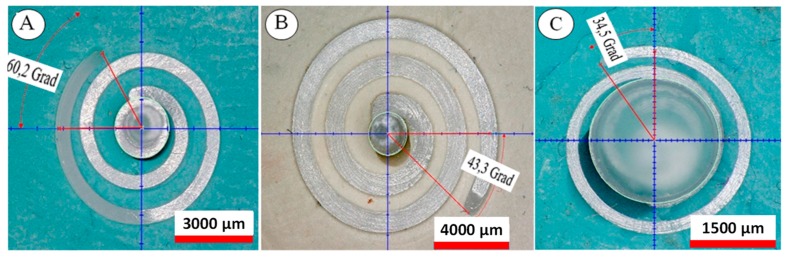
Example of measurements of the angle α for three different spiral parts with cross sections of (**A**) 0.5 × 0.5 mm^2^, flow length: 28.83 mm; (**B**) 1 × 1 mm^2^, flow length: 62.86 mm and (**C**) 0.25 × 0.25 mm^2^, flow length: 11.21 mm.

**Figure 5 micromachines-10-00335-f005:**
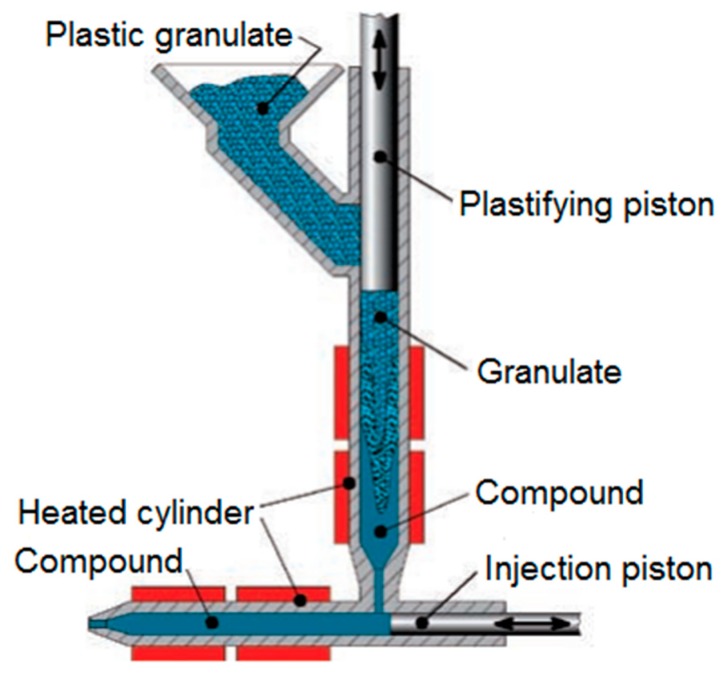
Working principle of FormicaPlast 2K [16].

**Figure 6 micromachines-10-00335-f006:**
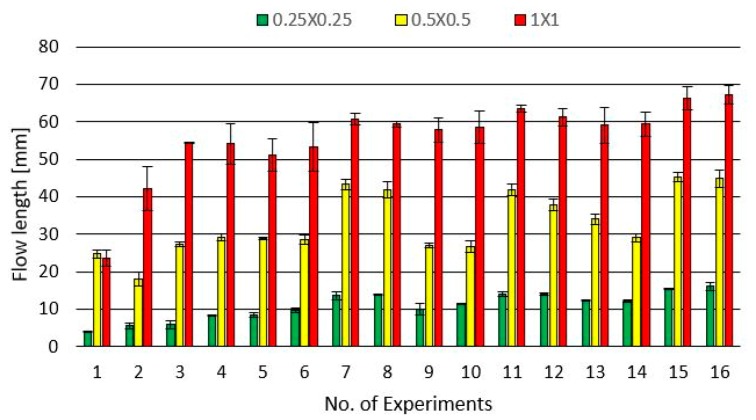
Achievable flow lengths for spiral geometries with different cross sections and for different process parameters.

**Figure 7 micromachines-10-00335-f007:**
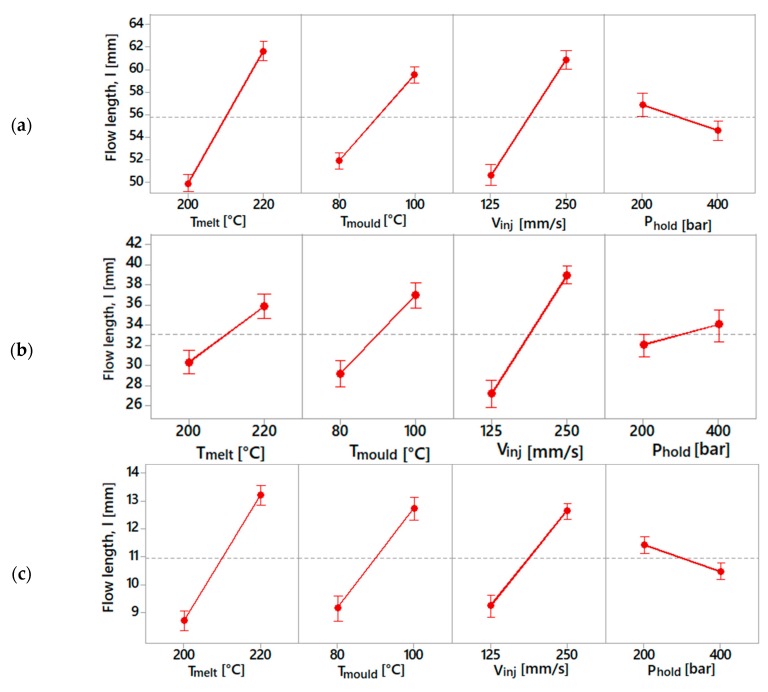
Main effect plot for spiral geometries with (**a**) 1 × 1 mm^2^, (**b**) 0.5 × 0.5 mm^2^ and (**c**) 0.25 × 0.25 mm^2^ cross sections.

**Figure 8 micromachines-10-00335-f008:**
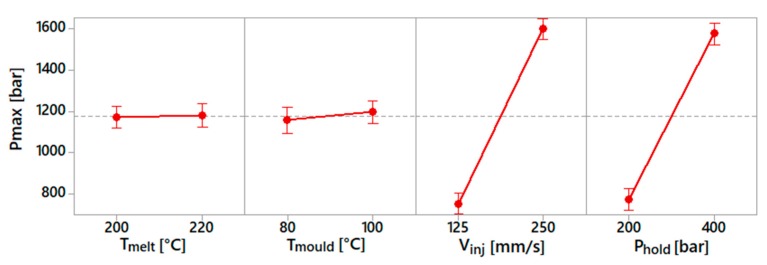
Main effect plot for maximum injection pressure.

**Figure 9 micromachines-10-00335-f009:**
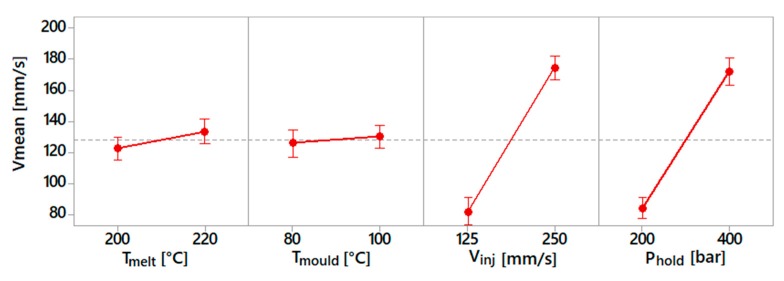
Main effect plot for mean injection speed.

**Figure 10 micromachines-10-00335-f010:**
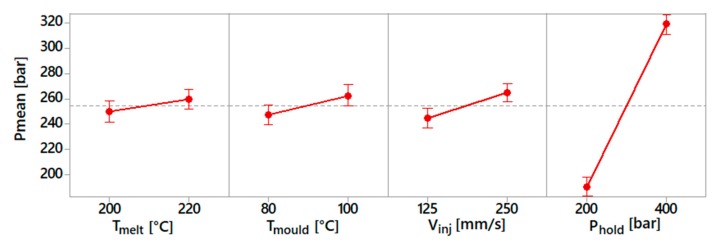
Main effect plot for mean injection pressure.

**Figure 11 micromachines-10-00335-f011:**
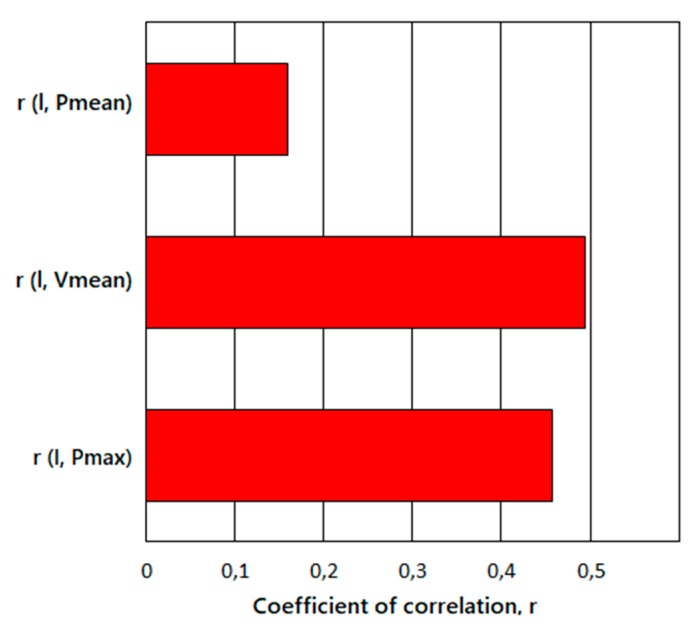
Values of the coefficient of correlation calculated between the flow length and the process fingerprints.

**Table 1 micromachines-10-00335-t001:** Values of the flow length for different radii for the spiral with a 1 × 1 mm^2^ cross section.

Quadrant	Radius (mm)	Max. Flow Length (mm)
0	1.5	1.5
1	2	4.63
2	2.5	8.55
3	3	13.25
4	3.5	18.75

**Table 2 micromachines-10-00335-t002:** Main properties of the material.

Property	Test Method	Units	Value
Density	ISO 1183	kg/m^3^	1400
Melt volume rate (190 °C, 2.16 kg)	ISO 1133	cm^3^/10 min	7.5
Melt temperature	ISO 11357-1/-3	°C	167

**Table 3 micromachines-10-00335-t003:** Process parameter settings for the design of experiments (DoE) plan.

Process Parameter	Low	High
Vinj (mm/s)	125	250
Tmelt (°C)	200	220
Tmould (°C)	80	100
Phold (bar)	200	400

**Table 4 micromachines-10-00335-t004:** Process parameter settings for the DoE plan.

Experiment	Tmelt (°C)	Tmould (°C)	Vinj (mm/s)	Phold (bar)
1	200	80	125	400
2	200	80	125	200
3	200	80	250	400
4	200	80	250	200
5	200	100	125	400
6	200	100	125	200
7	200	100	250	400
8	200	100	250	200
9	220	80	125	400
10	220	80	125	200
11	220	80	250	400
12	220	80	250	200
13	220	100	125	400
14	220	100	125	200
15	220	100	250	400
16	220	100	250	200

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
