# Peer review of "Analysis of the Downscaling Effect and Definition of the Process Fingerprints in Micro Injection of Spiral Geometries"

_micromachines, 2019, doi:10.3390/mi10050335_

Round 1
Reviewer 1 Report
In the opinion of the reviewer:
- It would be recommendable that the authors carefully proof read their paper
- 125 – 127 the authors said: The values are based on the material recommendations and preliminary experiments. In the opinion of the reviewer this is not enough. Results of screening experiments should be documented to understand the selection of the process parameter levels used in the DoE. Why the authors believe that only two level experiments of the investigate process parameter are sufficient to characterize the flow length of the molded polymer?
- Line 129-130 – what is the part functionality rather than flow length testing procedures?
- Figure 7 – the authors should include error bars for the different main effect plots of the experimental phase in order to evaluate the significant effect of the different process parameters investigated in this study.
- Figure 8, 9, 10 - the authors should include error bars for the different main effect plots to evaluate the significant effect of the different process parameters and their true sensitivity of the process indicators.
- Line 256 – 259 the authors stated: Finally, a correlation analysis was carried out to relate the response (i.e. achievable flow length) with the process fingerprints. By monitoring these process indicators, it will be possible to predict the response, avoiding time-consuming off-line measurements and performing an in-line quality assurance of the manufactured micro components. The reviewer don´t see a direct link of the highlighted findings with production of real products. If the approach for zero defect manufacturing is the focus of the current approach, the potential benefit of using it for different micro product geometries should be included and described in the paper.
Author Response
Thank you very much for your review. I've read carefully the paper and I improved the English language and the style.
Line 125 - 127: Preliminary experiments were carried out and about 100 components were manufactured by using different process parameters. Results of screening experiments are not show because they are not completely documented and because there will be too many results in the paper. Furthermore, most of the time results of preliminary experiments are not shown in papers. Two levels of experiments are enough to characterize the flow length since the main goal of the research is not to determine the effect of the process parameters on the achievable flow length. The main objective are to investigate the difference in flow length by downscaling the cross-section and to establish a relationship between the monitored process fingerprints and the process parameters.
Line 129 – 130: You are right. It is just a flow length testing procedure. Indeed I edited the sentence.
Figures 7, 8, 9, 10: I added the error bars, as requested.
Line 256 – 259: You are right, the explanation I gave is not clear. I added an explanation in the conclusions regarding the potential benefits of using this approach for different micro product geometries.
Reviewer 2 Report
Auto plagiarism found in the figure 5 with "Influence of variotherm injection moulding on the flow length of a micro part", Conference Paper · March 2017. Same authors. https://www.researchgate.net/publication/330422159_Influence_of_variotherm_injection_moulding_on_the_flow_length_of_a_micro_part
Parero chart should be shown as result of DoE
Author Response
Thank you very much for your review. I've read carefully the paper and I improved the English language and the style.
I do not agree with auto plagiarism for figure 5. The figure has been taken from the website of the injection moulding machine manufacturer and it has been modified by adding the writings. The website is clearly cited in the reference. The same image was used in a paper of mine but it is not a problem.
I prefer to do not show Pareto charts in this papers. It is already possible to see the factors that most affect the response by looking at the main effect plot diagrams. Furthermore, I added error bars in the main effect plot diagrams in order to give more accurate results.